# POINT CLOUD COMPLETION WITH LANDAU DISTRIBUTION: A PROBABILISTIC VIEW

## ABSTRACT

Point clouds are fundamental discrete representations used in computer vision, robotics, etc. Chamfer Distance (CD) is widely adopted as a metric and training loss to evaluate the similarity between two point clouds. However, the vanilla CD is sensitive to outliers, which means a few widely distributed points can disproportionately affect the final similarity score. Besides, CD calculates the simple average of distances of matched point pairs between two sets, which does not take into account the underlying point-wise distance distribution across two point clouds (same weights assigned for *short-* and *long*-distance pairs by using uniform distribution). To mitigate these issues, we analyze the effect of prioritizing short- and long-distance pairs with Gaussian distributions obtained with grid search, and based on the findings, we take an indirect approach to find Landau distribution, out of many distributions, fits in the form of bimodal Gaussian mixture model which balances two types of pairs. Based on this observation, we propose LandauCD, an innovative loss function grounded in the Landau distribution. We conduct comprehensive experiments using LandauCD and observe significant improvements consistently over all the popular baseline networks trained with CD-based losses, leading to new state-of-the-art results on several benchmarks (PCN, Shapet-55/34, ShapeNet-Part). We also delve into the theoretical explanation behind the consistent improvements of LandauCD. **Code and weights will be released upon acceptance.**

## 1 INTRODUCTION

**Point Cloud Completion.** Point clouds, which are straightforward to be collected using various sensing technologies, represent a cornerstone data format that has become increasingly important in the fields of modern robotics and automation. They are widely used in the tasks like object recognition, mapping, and navigation. (Wang et al., 2022; Ma et al., 2022; Shi et al., 2022). Nonetheless, it's important to note that the raw point cloud data collected by current 3D sensing technologies often suffers from incompleteness and sparsity. These limitations can arise from various factors such as occlusions, which block parts of the view, constrained sensor resolution, and issues related to light reflection or absorption. These challenges make the data less than ideal for immediate use in robotics and automation applications. (Yu et al., 2018; Li et al., 2021b; Luo & Hu, 2021; Li et al., 2021a; Zhou et al., 2022) and may adversely affect the efficacy of subsequent tasks requiring precise and high-quality data representations, such as point cloud segmentation and object detection. *Point cloud completion* (Alliegro et al., 2021) refers to the task of inferring the complete shape of an object or scene from incomplete raw point clouds. Recently, many (deep) learning-based approaches have been introduced to point cloud completion ranging from supervised learning and self-supervised learning to unsupervised learning (Yuan et al., 2018; Wang et al., 2020a; Mittal et al., 2021; Cai et al., 2022; Fan et al., 2022; Ren et al., 2022). Among these methods, supervised learning featuring a standard encoder-decoder architecture has emerged as the predominant choice for many researchers. This approach has been highly effective, setting new performance standards on almost all widely recognized benchmarks in the field of point completion. (Yu et al., 2021; Xiang et al., 2021; Zhou et al., 2022; Wang et al., 2022; Fei et al., 2022).

**Learning with Chamfer Distance (CD).** CD serves as a popular metric in the field of point cloud completion network design, such as SnowflakeNet(Xiang et al., 2021), PointAttN(Wang et al., 2022),

etc.(Guo et al., 2020; Wu et al., 2021). It evaluates the dissimilarities between any two point cloud sets by calculating the average distance of each point in one set to its nearest matching point in the other set. While CD can faithfully reflect the global dissimilarity by treating the distances of all nearest-neighbor pairs between both sets with equal importance. The formation of CD works as the uniform distribution weight operation for paired distance, and thus it is likely to be negatively affected by some points. Furthermore, by focusing on minimizing the Euclidean distances between paired points, it's commonly recognized that utilizing the CD for learning can be sensitive to outliers. As a consequence, this sensitivity to outliers often results in a phenomenon known as *clumping behavior*. In this scenario, a considerable number of points from one set correspond to a single point in another set, leading to the visual formation of small, dense clusters. This behavior can readily disrupt the commonly held assumption of uniform sampling from the underlying geometric surfaces, an assumption often used in the generation of point clouds, and thus makes the similarity measure of the underlying surfaces more challenging.

**Improved CD from Distribution View.** Intuitively, our view of understanding point clouds is considering it as the distribution of discrete sets of data points. Furthermore, in our point cloud completion task, we recognized the calculated CD also as a distribution, which elaborates the questions: *How can we accurately measure the similarity between two point clouds while incorporating their paired distances distribution data?* To answer this, we need to bring a loss function correlated with the paired distance distributions into discussion. An effective loss function should have two properties: **(1)** Capable of (re) weighting the paired distance distribution of the point sets dynamically with each paired object and **(2)** Considering information from points at both distant and near. From the formation of CD (Eq. 3), we know that the vanilla CD uses mean operation to weighted by a uniform distribution during the measurement of similarities. However, this method has limitations in learning: uniform distribution does not differentiate different distant data points when training, and there is no timely response to changes of paired distance distribution during the training process of different objects.

**Our Approach and Contributions.** Motivated by the distance distribution mentioned above, we seek a distribution function that can serve as a plug-in solution to re-weight distance pairs when used in optimizing as a replacement for the vanilla CD. Unlike the works in (Lin et al., 2023), which focus on emphasizing the optimization of well-matched point pairs (short-distance point pairs), we also aim to decrease the number of outliers (distant point pairs) while minimizing the distances of well-matched point pairs. Our approach is three-fold (Fig. 1) : **1)** We explore the effect of the shape of re-weighting functions on CD loss by comparing loss values under different settings on a small subset. Two Gaussian distributions are derived, which emphasize short- and long- distance point pairs, respectively. **2)** Considering these two Gaussian distributions as components, we test common distributions by calculating their dissimilarities (KL-divergence (Lyu et al., 2021)) to the two components. In this step, each chosen distribution function we used is first approximated as a weighted sum of two Gaussian distribution functions (Bhattacharya, 1967), which are then compared with the two components. **3)** The distribution function with the lowest dissimilarity is selected as the target re-weight function. LandauCD is thus proposed as a loss function by integrating Landau distribution into the vanilla CD. From the perspective of probability distributions, we can view LandauCD as an extension of vanilla CD. Conversely, the vanilla CD can be treated as a special case of re-weight loss functions, including LandauCD (when weights follow the uniform distribution).

To summarize, we list our main contributions as follows:

- We propose LandauCD by introducing Landau distribution into the CD loss, leading to a regularized CD loss which mitigates outliers while preserving the well-matched point pairs.
- We analyze the effect of re-weight function over distance distributions of point pairs and provide a systematic way of selecting a promising re-weighting function based on the a mechanism discussed in our work.
- We conduct comprehensive experiments for point cloud completion and achieve state-of-the-art results on popular benchmark datasets.

## 2 RELATED WORK

**Point Cloud Completion** PCN (Yuan et al., 2018), as the first learning-based point cloud completion network, extracts global features similarly to PointNet and generates points using FoldingNet's (Yang et al., 2018) folding operations. (Zhang et al., 2020) suggests extracting multi-scale features from different layers to capture local structures and improve performance. Attention mechanisms, notably the Transformer (Vaswani et al., 2017), excel at capturing long-range interactions, surpassing CNNs' limited receptive fields. SnowflakeNet (Xiang et al., 2021), PointTr (Yu et al., 2021) and SeedFormer (Zhou et al., 2022) accentuate the decoder component, incorporating Transformer designs. PointAttN (Wang et al., 2022), distinctly, is conceived entirely on Transformer foundations. These works have demonstrated the ability of Transformers in point cloud completion.

**Distance Metrics for Point Clouds.** Distance in point clouds is a non-negative function that measures the dissimilarity between them. With relatively low computational cost fair design, CD and its variants are extensively used in learning-based methods for point cloud completion tasks (Deng et al., 2019; Lyu et al., 2021; Zhang et al., 2022; Tang et al., 2022). Earth Mover's Distance (EMD), which is another widely used metric, relies on finding the optimal mapping function from one set to the other by solving an optimization problem. In some cases, it is considered to be more reliable than CD, but it suffers from high computational overhead and is only suitable for sets with exact numbers of points (Liu et al., 2020; Achlioptas et al., 2018). Recently, (Wu et al., 2021) propose a Density-aware Chamfer Distance (DCD) as a new metric for point cloud completion which can balance the behavior of CD and computational cost in EMD to a certain level.

**Landau Distribution.** In physics, ionization loss represents the energy loss in collision with target electrons when charged particles traverses matter (Fermi, 1940). To better simulate the fluctuation of the ionization energy loss, L.D. Landau employed the Laplace-Carson integral transform (Baerwald, 1936) and introduced dimensionless variable & parameters (Landau, 1944; Wilkinson, 1996; Grupen, 2000) on Bethe-Bloch formula (Bethe, 1933) to derive the Landau distribution as shown in Eq.1 under some ideal assumptions (Landau, 1944).

$$f_L(x) = \frac{1}{2\pi i} \int_{a-i\infty}^{a+i\infty} e^{s \log s + xs} ds \tag{1}$$

where $log(\cdot)$ represent the natural logarithm, $a \in \mathbb{R}^+$, $x$ is the dimensionless Landau's universal variable in (Wilkinson, 1996; Grupen, 2000; Bulyak & Shul'ga, 2022).

According to the approximation in Moyal et al. (Moyal, 1955), we can simplify Eq.1 to the following stable distribution:

$$p_L(x) = \frac{1}{\sqrt{2\pi}} \exp\left(-\frac{x + e^{-x}}{2}\right). \tag{2}$$

Landau distribution has been widely implemented on ionization loss calculation (Palmatier et al., 1955; Allison & Cobb, 1980; Nelson et al., 1985; Baró et al., 1995; Marucho et al., 2006) and random number generation (Schorr, 1973; Kölbig & Schorr, 1983). In our work, Landau distribution shown in Eq. 2 is utilized to constitute our novel loss function LandauCD.

## 3 METHODOLOGY

### 3.1 PRELIMINARIES

**Chamfer Distance Loss.** We denote $(x_i, y_i)$ as the $i$-th point cloud pair, with $x_i = \{x_{ij}\}$ and $y_i = \{y_{ik}\}$ as two sets of 3D points, and $d(\cdot, \cdot)$ as a certain distance metric. Then the CD loss for point clouds can be defined as follows:

$$\mathcal{L}_{\text{CD}}(x_i, y_i) = \ell_{\text{CD}}(x_i, y_i) + \ell_{\text{CD}}(y_i, x_i) = \frac{1}{|y_i|} \sum_k \min_j d(x_{ij}, y_{ik}) + \frac{1}{|x_i|} \sum_j \min_k d(x_{ij}, y_{ik}),$$

$$\tag{3}$$

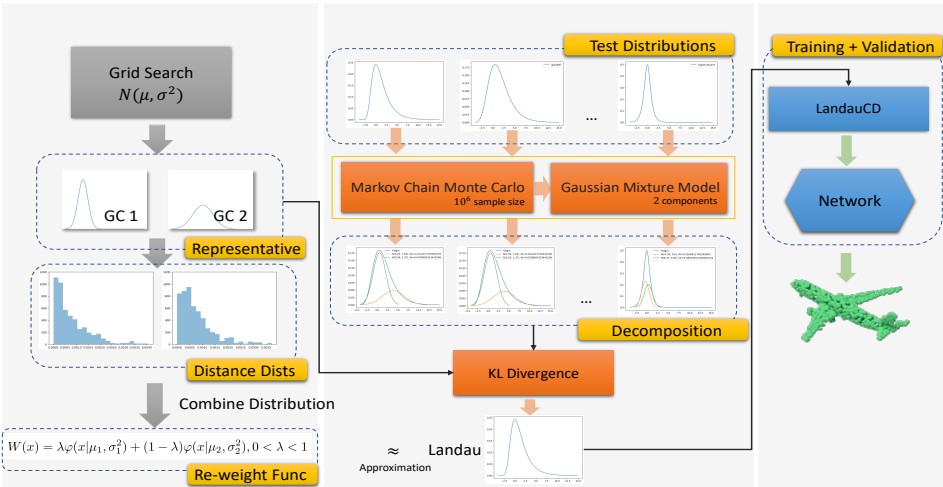

Figure 1: Flowchart of Optimal Distribution-Based Loss Function Searching Mechanism

For point cloud completion, function $d$ usually refers to Euclidean $\ell_1$ or $\ell_2$ norm of a vector.

**Challenges and Motivations.** CD (Eq. 3) calculates the averaged value of matched point pair distances, which is equivalent to uniformly re-weighting all the distances of matched point pairs. As a classic issue of averaging, the value of CD can be disproportionally impacted by outliers (matched point pairs with high distances). The vulnerability of CD to outliers is also discussed in (Wu et al., 2021; Lin et al., 2023), which can lead to the drift towards suboptimal models. To address this issue, different re-weighting mechanisms have been applied in the above-mentioned works. However, most of the re-weighting mechanisms treat distances of point pairs as a whole without considering characteristics (the distribution of distances) of point pairs. Furthermore, these re-weighting mechanisms tend to de-prioritize the outliers while over-weighting short-distance pairs, which does not limit the number of outliers. In the light of the previous works, we hope to find a re-weighting mechanism which is capable of balancing the short- and long-distance point pairs. More specifically, an ideal weighting mechanism in our perspectives should be able to preserve/improve the quality of short-distance point pairs while limiting the number of long-distance point pairs (as a way to reduce outliers).

### 3.2 ANALYSIS

**Prioritizing Point Pairs with Different Distances.** Previous study(Lin et al., 2023) indicates that prioritizing short-distance point pairs benefits the completion task. This naturally led us to ponder: *would prioritizing long-distance point pairs also help the training process?* To answer this question, we first use grid search and find two Gaussian distributions. Gaussian is chosen because of its relatively small search space (it can fully characterized with only two parameters). After conducting a grid search, two representative Gaussian distribution ($\mathcal{N}(\mu_1, \sigma_1^2)$ and $\mathcal{N}(\mu_2, \sigma_2^2)$) are selected (these two distributions are shown in Fig. 3 (b) as **GC 1** and **GC 2**). These two distributions are then used independently as re-weight functions in training. The experiment results align with our hypothesis. Both Gaussian distributions yield promising results, which implies prioritizing long-distance point pairs can also lead to performance similar to prioritizing short-distance point pairs.

Albeit the similarity in performances, the difference in using two components independently can be easily seen in the comparison of both distributions of distances in matched point pairs (**Fig.** 2). By prioritizing the short-distance point pairs (**GC 1**), we observe more point pairs with very small distances ($0.5 \times 10^{-3}$). On the other hand, by prioritizing the long-distance pairs (**GC 2**), the portion of pairs with longer distances ($4 \times 10^{-3}$) is reduced. The same pattern is found in different epochs of training. These observations also prove prioritizing long-distance point pairs is a valid way to reduce outliers.

**Combining Distributions.** To leverage the optimization effects of both Gaussian components, a natural and simple way of thinking is to combine these two components into one function with a

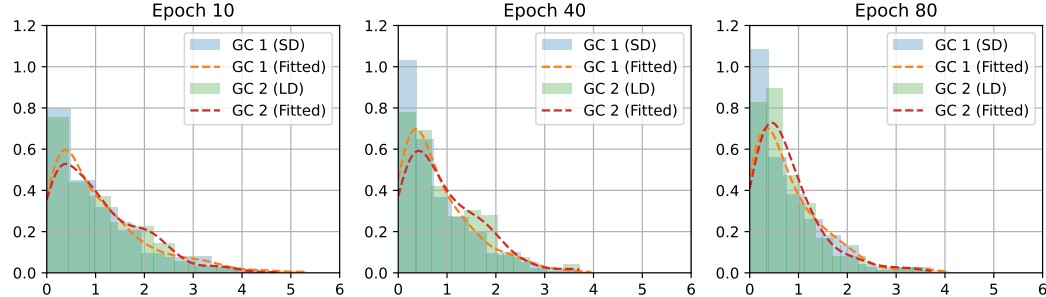

Figure 2: Comparisons of Distributions of Distances in Matched Point Pairs (Distances (x axis) are multiplied by $10^3$ in visualization). The three plots corresponds to distance distribution sampled at different training epochs. The representative Short-Distance **GC 1** and Long-Distance **GC 2** components are used independently as re-weight function in training.

weighted summation. The combination of both components can have some characteristics from both sides, as mentioned in the previous section. Expressed in mathematical form, the re-weight function should take the following form.

$$W(x) = \lambda\varphi(x|\mu_1, \sigma_1^2) + (1 - \lambda)\varphi(x|\mu_2, \sigma_2^2), 0 < \lambda < 1 \tag{4}$$

We use $\lambda$ as the factor to balance the two Gaussian components (denoted as $\varphi(x|., .)$) and $\mu_i, \sigma_i^2$ are the mean and variance of a given component. It is easy to tell the re-weight function takes the form of a bimodal Gaussian mixture model (Reynolds et al., 2009).

Thus, the CD loss function with the re-weight function should take the following form.

$$CD_W(S_1, S_2) = \sum_{x \in S_1} \hat{y}W(\hat{y}) + \sum_{y \in S_2} \hat{x}W(\hat{x}), \tag{5}$$

where $\hat{y} = \min_{y \in S_2} ||x - y||_2, \hat{x} = \min_{x \in S_1} ||y - x||_2$.

**Smooth Curve Constraint.** The re-weight function adopts the form of Gaussian mixture model, however, the choice of $\lambda$ is not arbitrary. As a bimodal Gaussian mixture model, most of selections of $\lambda$ will result in two visible peaks in the visualization of function, creating troughs of valleys between them. These configurations lead to sudden changes or unnecessary sinuous fluctuations in the shape of the re-weight function, which might cause point pairs with similar distances receiving disproportionate weights when used in training.

The issue of configurations in bimodal Gaussian mixture model becomes more evident in our setting. The two Gaussian components characterized by $\mu_1, \sigma_1$ and $\mu_2, \sigma_2$ are not located near each other (in other words, the absolute value of $\mu_1 - \mu_2$ is usually not negligible), since these two components are prioritizing different parts in the spectrum of distance distribution of point pairs. As a result, there is not a straightforward way of choosing a $\lambda$ with the constraint satisfied.

**Indirect Solution to the Constrained Problem.** While the choice of $\lambda$ is difficult, we opt to take an indirect approach to solve a similar question which is more manageable: *which common distribution function can be approximated as a bimodal Gaussian mixture model while both split-ted distributions are close to the Gaussian components mentioned above?*

The intuition behind this approach is straightforward: most of the common distribution functions are smooth in nature. To avoid excessively large search space introduced by additional parameters in common distributions, we limit values parameters in each test distributions to take its most simplistic form. The common distributions and the parameter configurations are listed in Table 4.

To approximate each of the tested distribution $p$ as a bimodal Gaussian mixture model, we apply Markov Chain Monte Carlo (MCMC) method(Brooks, 1998) to firstly sample from each distribution. We keep $10^6$ sample size for each of the test distribution. The sample is then fed into a Gaussian Mixture Model to generate two components. The two components ($f_1$ and $f_2$) take the following form, which is similar to Eq. 4. Note $\lambda^{'}$ is automatically determined in this process. For example, Fig. 3 (a) shows the two Gaussian components generated by applying Gaussian Mixture Model decomposition on Landau distribution.

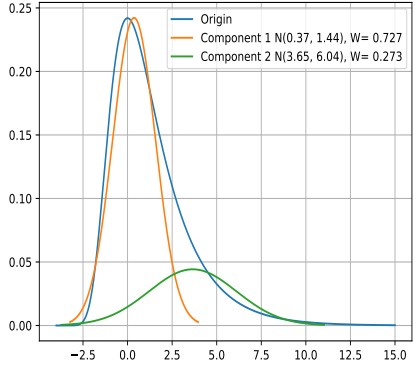

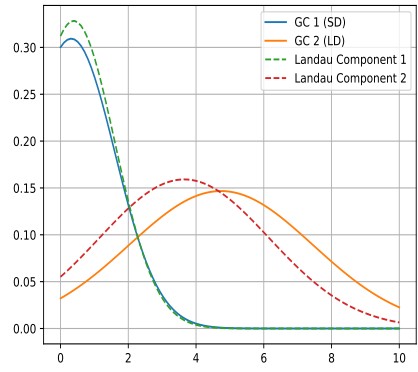

(a) Decomposition of Landau Distribution (MCMC +GMM )

(b) Representative Gaussian Components and Landau Decomposition Results

Figure 3: (a) Decomposition of Landau Distribution. Landau distribution is approximated with the weighted summation of two Gaussian components. (b) The comparison between the representative Gaussian components (GC 1 and GC 2) and the components obtained by GMM decomposition.

$$p(x|...) = \lambda^{'} \varphi(x|\mu_1^{'}, \sigma_1^{'2}) + (1 - \lambda^{'}) \varphi(x|\mu_2^{'}, \sigma_2^{'2}), 0 < \lambda^{'} < 1 \tag{6}$$

**Evaluating the Two Components.** We compare the two components with the components obtained from grid search using KL divergence. This operation is done for all the test distributions. The test distribution with the smallest KL divergence is selected as the re-weight function. This operation can be expressed as:

$$W \approx \tilde{W} = \underset{p \in \mathbf{P}}{\operatorname{argmin}} \mathbf{KL}\left(\mathcal{N}(\mu_1^{'}, \sigma_1^{'2}), \mathcal{N}(\mu_1, \sigma_1^2)\right) + \mathbf{KL}\left(\mathcal{N}(\mu_2^{'}, \sigma_2^{'2}), \mathcal{N}(\mu_2, \sigma_2^2)\right) \tag{7}$$

where $p$ is a single test distribution of all the test distributions $\mathbf{P}$ we experiment with. The values of two split-ted components ($\mu_1^{'}, \sigma_1^{'}, \mu_2^{'}$ and $\sigma_2^{'}$) are obtained from decomposition results with the Gaussian Mixture Model. We assume $\mu_1 < \mu_2$ and $\mu_1^{'} < \mu_2^{'}$ for all the comparisons. The KL divergence values for all the test distributions we choose are tabulated in Table 4. We can see among all the test distributions, the Landau distribution has the minimal KL divergence score, which indicates it is the most similar approximation to Eq. 4. In other words, Landau is selected as $\tilde{W}$ as an approximation of $W$. We can also view Landau as one of the smooth functions that are capable of balancing the priorities of short-distance and long-distance during training. In Fig. 3 (b), we can see the decomposition results of Landau distribution and two representative components are also visually close.

Evaluating using $W$ (the value of $\lambda$ is set to $\lambda^{'}$) and $\tilde{W}$ yields similar results (CD on $W$: 4.03; CD on $\tilde{W}$: 4.00), which both outperform models trained with a single Gaussian component alone by a large margin (CD on GC1: 4.08; CD on GC2: 4.07). $\tilde{W}$ (Landau) gives a slightly better result compared with $W$, which can be explained by the higher smoothing nature in Landau.

### 3.3 LANDAUCD LOSS

Based on the analysis, we propose LandauCD as a loss function by integrating probability distribution of Landau $p_{Landau}(x)$ into vanilla CD (we substitute $W$ with $p_{Landau}(x)$ in Eq. 5), and the loss function is shown as follows.

$$CD_{Landau}(S_1, S_2) = \sum_{x \in S_1} \hat{y} \frac{1}{\sqrt{2\pi}} \exp\left(-\frac{\hat{y} + e^{-\hat{y}}}{2}\right) + \sum_{y \in S_2} \hat{x} \frac{1}{\sqrt{2\pi}} \exp\left(-\frac{\hat{x} + e^{-\hat{x}}}{2}\right), \tag{8}$$

where $\hat{y} = \min_{y \in S_2} ||x - y||_2, \hat{x} = \min_{x \in S_1} ||y - x||_2$.

## 4 EXPERIMENTS

**Datasets.** We conduct experiments for point cloud completion on the following benchmark datasets:

- *PCN (Yuan et al., 2018):* This dataset is a subset of ShapeNet (Chang et al., 2015), encompassing shapes from eight categories. The incomplete point clouds are derived by back-projecting 2.5D depth images from eight viewpoints, mimicking real-world sensor data. Each shape has 16,384 points uniformly sampled from mesh surfaces as complete ground truth, with 2,048 points sampled as partial input (Tchapmi et al., 2019; Zhou et al., 2022).
- *ShapeNet-55/34 (Yu et al., 2021):* ShapeNet-55 contains 55 categories in ShapeNet, with 41,952 shapes for training and 10,518 shapes for testing. ShapeNet-34 uses a subset of 34 categories for training and leaves 21 unseen categories for testing, where 46,765 object shapes are used for training, 3,400 for testing on seen categories, and 2,305 for testing on novel (unseen) categories. In both datasets, 2,048 points are sampled as input and 8,192 points as ground truth. Following the same evaluation strategy with (Yu et al., 2021), 8 fixed viewpoints are selected and the number of points in the partial point cloud is set to 2,048, 4,096 or 6,144 (25%, 50% or 75% of a complete point cloud) which corresponds to three difficulty levels of *simple*, *moderate* and *hard* in the test stage.
- *ShapeNet-Part (Yi et al., 2016):* This dataset is a subset of ShapeNetCore 3D meshes, encompassing 17,775 distinct 3D meshes across 16 categories. The ground-truth point clouds are obtained by uniformly sampling 2,048 points on each mesh. The partial point clouds are generated by randomly selecting a viewpoint from multiple viewpoints as a center and eliminating points within a specified radius from the complete data, with a total of 512 points being removed from each point cloud.

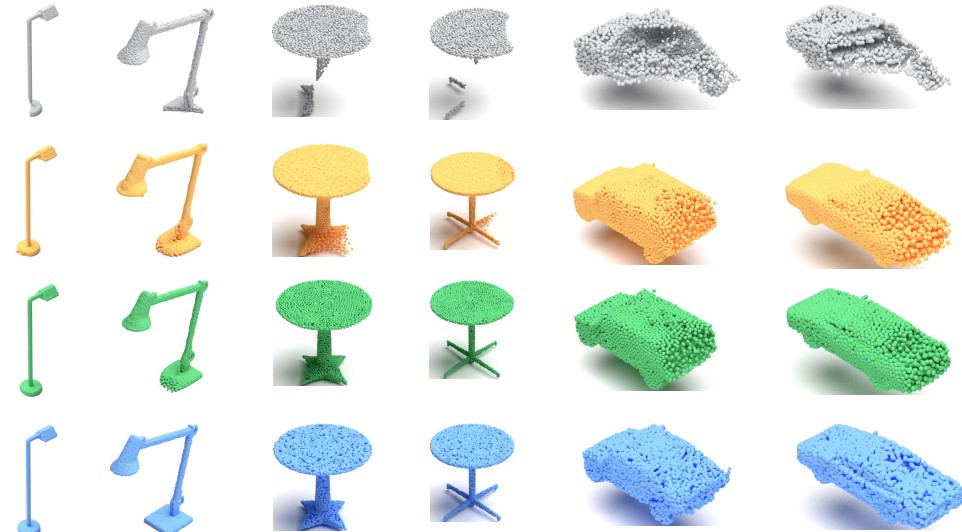

Figure 4: Visual Comparison on PCN. **Row-1:** Inputs of Incomplete Point Clouds. **Row-2:** Outputs of Seedformer with CD. **Row-3:** Outputs of Seedformer with LandauCD. **Row-4:** Ground truth.

**Implementation.** We first take three state-of-the-art networks, CP-Net (Lin et al., 2022), PointAttN (Wang et al., 2022) and SeedFormer (Zhou et al., 2022), as our backbone networks for comparison and analysis. We also apply LandauCD to almost all the popular completion networks in recent years, FoldingNet (Yang et al., 2018), PMP-Net (Wen et al., 2021), PoinTr (Yu et al., 2021), SnowflakeNet (Xiang et al., 2021), to verify its performance by replacing the original CD loss wherever it occurs. We do the same replacement for all the other comparative losses in our experiments. We train all these networks from scratch using PyTorch, optimized by either Adam (Kingma & Ba, 2014) or AdamW (Loshchilov & Hutter, 2017). To ensure fairness in comparison, we replace the loss functions from all the stages with LandauCD so it can participate in the whole training process. Hyperparameters such as learning rates, batch sizes and balance factors in the original losses for training baseline networks are kept consistent with the baseline settings. We conduct our experiments on a server with

4 NVIDIA A100 80G GPUs and one with 10 NVIDIA Quadro RTX 6000 24G GPUs due to the large model sizes of some baseline networks.

**Evaluation.** Following the literature, we evaluate the best performance of all the methods using CD (lower is better). We also use F1-Score@1% (Tatarchenko et al., 2019) (higher is better) to evaluate the performance on ShapeNet-55/34. For better comparison, we cite the original results of some other methods on PCN and ShapeNet-55/34.

## 4.1 STATE-OF-THE-ART COMPARISON

**PCN.** In accordance with the literature, we report the CD with L1-distance in Table 1. As we can see, LandauCD significantly enhances the performance across all baselines consistently, achieving new state-of-the-art results. As previously mentioned, numerical metrics like CD may not accurately encapsulate the visual quality; hence, we also furnish qualitative evaluation results in Fig. 4, juxtaposed with outcomes generated from Seedformer trained with CD loss. It is discernible that both models can approximate point clouds in general outlines to a certain degree, yet the completion results em-

Table 1: Comparison on PCN in terms of per-point L1-CD ×1000.

| Methods | Plane | Cabinet | Car | Chair | Lamp | Couch | Table | Boat | Avg. |
|---|---|---|---|---|---|---|---|---|---|
| TopNet (Tchapmi et al., 2019) | 7.61 | 13.31 | 10.90 | 13.82 | 14.44 | 14.78 | 11.22 | 11.12 | 12.15 |
| AtlasNet (Groueix et al., 2018) | 6.37 | 11.94 | 10.10 | 12.06 | 12.37 | 12.99 | 10.33 | 10.61 | 10.85 |
| GRNet (Xie et al., 2020) | 6.45 | 10.37 | 9.45 | 9.41 | 7.96 | 10.51 | 8.44 | 8.04 | 8.83 |
| CRN (Wang et al., 2020b) | 4.79 | 9.97 | 8.31 | 9.49 | 8.94 | 10.69 | 7.81 | 8.05 | 8.51 |
| NSFA (Zhang et al., 2020) | 4.76 | 10.18 | 8.63 | 8.53 | 7.03 | 10.53 | 7.35 | 7.48 | 8.06 |
| FBNet (Yan et al., 2022) | 3.99 | 9.05 | 7.90 | 7.38 | 5.82 | 8.85 | 6.35 | 6.18 | 6.94 |
| PCN (Yuan et al., 2018) | 5.50 | 22.70 | 10.63 | 8.70 | 11.00 | 11.34 | 11.68 | 8.59 | 11.27 |
| FoldingNet (Yang et al., 2018) | 9.49 | 15.80 | 12.61 | 15.55 | 16.41 | 15.97 | 13.65 | 14.99 | 14.31 |
| **LandauCD+FoldingNet** | **7.30** | **12.69** | **10.46** | **13.00** | **11.92** | **13.39** | **10.86** | **10.59** | **11.27** |
| PMP-Net (Wen et al., 2021) | 5.65 | 11.24 | 9.64 | 9.51 | 6.95 | 10.83 | 8.72 | 7.25 | 8.73 |
| **LandauCD+PMP-Net** | **4.59** | **10.10** | **8.90** | **8.57** | **6.38** | **10.47** | **7.49** | **6.75** | **7.92** |
| PoinTr (Yu et al., 2021) | 4.75 | 10.47 | 8.68 | 9.39 | 7.75 | 10.93 | 7.78 | 7.29 | 8.38 |
| **LandauCD+PoinTr** | **4.12** | **9.49** | **8.07** | **7.82** | **6.30** | **9.28** | **6.76** | **6.41** | **7.28** |
| SnowflakeNet (Xiang et al., 2021) | 4.29 | 9.16 | 8.08 | 7.89 | 6.07 | 9.23 | 6.55 | 6.40 | 7.21 |
| **LandauCD+SnowflakeNet** | **3.98** | **8.97** | **7.78** | **7.40** | **5.76** | **8.86** | **6.16** | **6.14** | **6.88** |
| PointAttN (Wang et al., 2022) | 3.87 | 9.00 | 7.63 | 7.43 | 5.90 | 8.68 | 6.32 | 6.09 | 6.86 |
| **LandauCD+PointAttN** | **3.72** | **8.88** | **7.46** | **7.04** | **5.60** | **8.47** | **6.24** | **5.93** | **6.66** |
| SeedFormer (Zhou et al., 2022) | 3.85 | 9.05 | 8.06 | 7.06 | 5.21 | 8.85 | 6.05 | 5.85 | 6.74 |
| **LandauCD+SeedFormer** | **3.65** | **8.68** | **7.64** | **6.80** | **5.04** | **8.57** | **5.79** | **5.71** | **6.49** |

ploying CD are prone to distortion in several regions with high surface noise levels. Conversely, LandauCD significantly aids the baseline network in better reconstructing point clouds within general outlines while preserving the realistic details of the original ground truth and effecting a notable reduction in noise.

**ShapeNet-55/34.** We assess the adaptability of LandauCD across both datasets for tasks with higher diversities. Table 2 enumerates the L2-CD across three levels of difficulty, alongside the average. In accordance with the literature, we delineate the results across five categories (Table, Chair, Plane, Car, and Sofa) which have training sample counts exceeding 2,500, as presented in the table. Addi-

Table 2: Results on ShapeNet-55 using L2-CD×1000 and F1 score.

| Methods | Table | Chair | Plane | Car | Sofa | CD-S | CD-M | CD-H | Avg. | F1 |
|---|---|---|---|---|---|---|---|---|---|---|
| PFNet | 3.95 | 4.24 | 1.81 | 2.53 | 3.34 | 3.83 | 3.87 | 7.97 | 5.22 | 0.339 |
| TopNet | 2.21 | 2.53 | 1.14 | 2.18 | 2.36 | 2.26 | 2.16 | 4.3 | 2.91 | 0.126 |
| PCN | 2.13 | 2.29 | 1.02 | 1.85 | 2.06 | 1.94 | 1.96 | 4.08 | 2.66 | 0.133 |
| GRNet | 1.63 | 1.88 | 1.02 | 1.64 | 1.72 | 1.35 | 1.71 | 2.85 | 1.97 | 0.238 |
| FoldingNet | 2.53 | 2.81 | 1.43 | 1.98 | 2.48 | 2.67 | 2.66 | 4.05 | 3.12 | 0.082 |
| **LandauCD+F.** | **2.09** | **2.32** | **1.01** | **1.50** | **2.01** | **2.15** | **2.46** | **3.39** | **2.66** | **0.141** |
| PoinTr | 0.81 | 0.95 | 0.44 | 0.91 | 0.79 | 0.58 | 0.88 | 1.79 | 1.09 | 0.464 |
| **LandauCD+P.** | **0.69** | **0.83** | **0.33** | **0.80** | **0.67** | **0.43** | **0.70** | **1.47** | **0.88** | **0.527** |
| SeedFormer | 0.72 | 0.81 | 0.40 | 0.89 | 0.71 | 0.50 | 0.77 | 1.49 | 0.92 | 0.472 |
| **LandauCD+S.** | **0.67** | **0.73** | **0.34** | **0.82** | **0.62** | **0.45** | **0.73** | **1.39** | **0.86** | **0.489** |

tionally, we furnish the results utilizing the F1 metric. Yet again, LandauCD has markedly enhanced the baseline models, particularly in instances where networks are simpler, such as FoldingNet.

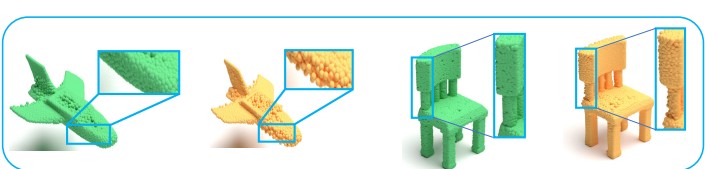

Figure 5: Detailed Visual on Seedformer. Green with LandauCD. Yellow with CD.

On ShapeNet-34, we assess performances within 34 seen categories (identical to training) as well as 21 unseen categories (not utilized in training), and enumerate our results in Table 3. It is observable that, once again, LandauCD enhances the performance of baseline models, suggesting that LandauCD is highly generalizable for point cloud completion tasks. We also provide some details results in Fig. 5.

Table 3: Results on ShapeNet-34 using L2-CD×1000 and F1 score.

| Methods | 34 seen categories | | | | | 21 unseen categories | | | | |
|---|---|---|---|---|---|---|---|---|---|---|
| | CD-S | CD-M | CD-H | Avg. | F1 | CD-S | CD-M | CD-H | Avg. | F1 |
| PFNet | 3.16 | 3.19 | 7.71 | 4.68 | 0.347 | 5.29 | 5.87 | 13.33 | 8.16 | 0.322 |
| TopNet | 1.77 | 1.61 | 3.54 | 2.31 | 0.171 | 2.62 | 2.43 | 5.44 | 3.50 | 0.121 |
| PCN | 1.87 | 1.81 | 2.97 | 2.22 | 0.154 | 3.17 | 3.08 | 5.29 | 3.85 | 0.101 |
| GRNet | 1.26 | 1.39 | 2.57 | 1.74 | 0.251 | 1.85 | 2.25 | 4.87 | 2.99 | 0.216 |
| FoldingNet | 1.86 | 1.81 | 3.38 | 2.35 | 0.139 | 2.76 | 2.74 | 5.36 | 3.62 | 0.095 |
| **LandauCD+F.** | **1.50** | **1.57** | **3.04** | **2.03** | **0.174** | **2.40** | **2.45** | **5.02** | **3.29** | **0.154** |
| PoinTr | 0.76 | 1.05 | 1.88 | 1.23 | 0.421 | 1.04 | 1.67 | 3.44 | 2.05 | 0.384 |
| **LandauCD+P.** | **0.44** | **0.65** | **1.29** | **0.79** | **0.525** | **0.63** | **1.07** | **2.54** | **1.41** | **0.492** |
| SeedFormer | 0.48 | 0.70 | 1.30 | 0.83 | 0.452 | 0.61 | 1.08 | 2.37 | 1.35 | 0.402 |
| **LandauCD+S.** | **0.42** | **0.64** | **1.23** | **0.76** | **0.580** | **0.56** | **1.03** | **2.17** | **1.25** | **0.447** |

Table 4: CP-Net Avg. Results on ShapeNet-Part.

| Distribution | Parameters | Bimodal $\mathcal{N} \sim (\mu, \sigma^2)$ | KL-Divergence | L2-CD×$10^3$ |
|---|---|---|---|---|
| Levy | $\mu$=0.0, $\beta$=0.5 | (4.670, 12.242), (0.494, 0.189) | 15.68 | 4.83 |
| Hyper secant | $\mu$=0.0, $\beta$=0.0 | (0.037, 0.934), (-0.344, 0.79) | 1.11 | 4.74 |
| Laplace | $\mu$=0.0, $\beta$=1.0 | (-0.035, 0.771), (0.254, 4.078) | 0.41 | 4.55 |
| Cauchy | $\mu$=0.0, $\beta$=0.5 | (-0.062, 1.849), (5.329, 12.955) | 0.26 | 4.15 |
| Gumbel | $\mu$=0.5, $\beta$=2.0 | (0.578, 2.388), (4.003, 7.134) | 0.09 | 4.19 |
| Landau | $\mu$=0.0, $\beta$=1.0 | (0.375, 1.437), (3.613, 6.069) | **0.03** | **4.00** |

**Analysis.**

We have chosen the ShapeNet-Part dataset for analysis, in comparison with various distribution-based loss functions. As previously introduced, ShapeNet-Part is a relatively compact dataset encompassing 16 categorical objects, which suffices for the analysis in our case. Regarding the model architecture, we have selected a lightweight network, denoted as CP-Net(Lin et al., 2022). With CP-Net on ShapeNet-Part dataset, we conduct intensive experiments for verifying our hypothesis and analyzing our results, Table 4 provides the common distribution behavior and its related hyper-parameters. We also summarize our result in Table 5 and compare with the results of CP-Net trained with some popular loss functions.

Table 5: CP-Net Results on ShapeNet-Part.

| Loss | L2-CD×$10^3$ |
|---|---|
| L1-CD | 4.16 |
| L2-CD | 4.82 |
| DCD (Wu et al., 2021) | 5.74 |
| **LandauCD** | **4.00** |

## 5 CONCLUSION

We propose a new loss function for point cloud completion, namely LandauCD, which re-weights the CD loss and prioritizes alignment the paired distance distributions between prediction and ground truth from both short- and long-distance point pairs. In particular, we discuss and analyze common probabilistic distributions and select Landau as our optimal solution. Comprehensive experiments have been conducted to demonstrate its effectiveness and efficiency using 7 networks on 4 datasets, leading to new state-of-the-art results.

**Limitations.** While we observe prioritizing point pairs with single Gaussian distribution can lead to expected behaviors (distance distribution changes), the optimization behavior of combining two distributions with weighted summation still needs more rigorous investigation. Besides, the use of grid search in our analysis limits the granularity of parameters when searching for representative distributions for short- and long-distance point pairs.

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
