# OpenReview forum: "Point Cloud Completion with Landau Distribution: A Probabilistic View"
_ICLR.cc/2024/Conference — ICLR 2024 Conference Withdrawn Submission_

### Official Review · Reviewer_oZ1H · 2023-10-30

**Soundness:** 3 good
**Presentation:** 1 poor
**Contribution:** 2 fair
**Rating:** 6
**Confidence:** 3

**Summary:**

The paper addresses the issue of using Chamfer Distance (CD) as a metric for point cloud completion, known for its sensitivity to outliers. The paper introduces Landau Chamfer Distance, leveraging the Landau distribution to better weight short and long-distance pairs in the loss function. Experimental results reveal consistent improvements in point cloud completion quality when replacing traditional CD with Landau CD. This approach offers enhanced robustness to outliers, promising advancements in point cloud completion accuracy for applications like 3D object reconstruction and computer vision.

**Strengths:**

1. The paper's focus on improving the standard Chamfer Distance is highly relevant, as CD is a critical and widely used loss function for optimization point cloud completion models. Enhancing its performance is crucial for various applications in 3D object reconstruction and computer vision.

2. The paper employs a rigorous evaluation methodology by exclusively replacing the loss function during training. This approach effectively isolates the effects of the proposed method, allowing for a clear assessment of its impact on point cloud completion. This meticulous testing strategy ensures the paper's contributions are thoroughly validated.

3. The paper provides quantitative results that consistently showcase improvements in point cloud completion quality when utilizing the proposed Landau Chamfer Distance. These consistent findings reinforce the efficacy of the approach, suggesting its reliability and potential for practical use.

4. The paper's evaluations encompass multiple models and datasets, demonstrating the versatility and adaptability of the proposed method. By testing on diverse scenarios, the paper underscores the generalizability of the Landau Chamfer Distance approach, making it applicable to a wide range of point cloud completion tasks and data sources. This comprehensive evaluation strengthens the paper's contribution and its potential impact in various domains.

**Weaknesses:**

Weaknesses Identified in the Paper:

1. The paper suffers from poor clarity, especially in the method section. The writing should be simplified and clarified to ensure that the proposed approach and its rationale are more easily understood, even by people who are not very familiar with the topic.

2. The initial sections of the introduction are structured more like the beginning of a related work section, and the introduction should primarily focus on why the proposed approach is necessary rather than reviewing the literature on point cloud completion. The paper's potential impact beyond point cloud completion should be emphasized in the introduction, making it a motivating "funnel."
Merging the first two sections of the introduction into the related work section would help streamline the paper and improve its flow.

3. The method section requires improvement in its description. The challenges and motivation could be merged into the introduction. In the method section, the paper should first provide a high-level overview of the proposed approach before justifying each component's selection, such as the choice of grid search and the distance function.

4. The analysis section and Figure 2 need enhancement. For example, the paper mentions that differences between the two components can easily be seen in Figure 2. This might be the case for people who are very familiar with the topic, but to me, the Figure did not clearly show any differences of particular significance. I suggest being more specific in making the arguments in the analysis part. This should include improvements to Figure 2, maybe by highlighting the most important differences.

5. The language used in the paper can be imprecise. For instance, stating that a pattern across different epochs "proves" the validity of prioritizing long-distance pairs to reduce outliers is misleading; instead, the paper should clarify that it provides evidence in support of this idea.

6. The qualitative results presented in Figure 4 appear to show marginal improvements at best. It would be beneficial to include more representative examples that clearly demonstrate the benefits of the proposed loss function qualitatively. Alternatively, the paper may use zoomed-in sections and highlighting, similar to Figure 5, to provide a more illustrative presentation of the results.

**Questions:**

Overall, I think that the paper addresses an important problem in the literature, provides a compelling framework for evaluation, and shows consistent improvements. However, in my opinion, the paper requires significant polishing in its writing and presentation to meet the standards for publication. Addressing these points during the rebuttal process could lead to an improved manuscript and a potentially higher score from my end.

---

### Official Review · Reviewer_eHnS · 2023-10-31

**Soundness:** 2 fair
**Presentation:** 2 fair
**Contribution:** 2 fair
**Rating:** 3
**Confidence:** 5

**Summary:**

This paper examines the limitations of the Chamfer Distance and introduces a new distance metric called LandauCD by incorporating the Landau Distribution. The proposed metric converts point clouds into distributions and measures the disparity between these distributions as the difference between the point clouds. However, the similarity to existing methods restricts its novelty. The theoretical analysis presented in the paper is challenging to comprehend. In terms of the experiments, the absence of a comparison with existing methods diminishes the clarity of the proposed method's superiority.

**Strengths:**

This paper measures the distance between point clouds in a distribution view, where many theorems could be used to analyze the problem.

**Weaknesses:**

1) The novelty of the paper is limited since the formula of the proposed distance metric is similar to the previous works. Besides, the theoretical foundation presented in the manuscript is difficult to read, making it difficult to understand the motivation of the proposed distance metric.

2)The experiments lack a comparison with existing methods, including both quantitative and visual results.

**Questions:**

1)	In Section 3.2, Paragraph 1, it is mentioned that the grid search and two Gaussian distributions are used to validate whether prioritizing long-distance point pairs also helps the training process. However, the authors do not explain the relationship between these two distributions and the prioritization of long-distance and short-distance point pairs.

2)	The proposed distance metric is similar to the previous ones, such as DCD [1], where the only difference is the form of the weights. I think the authors should describe the insight differences between them.

3)	In Eq. (3), what is l_{CD}? In Eq. (5), what is the definition of S_1 and S_2?

4)	In Sec. 3.1, the authors claim that the outliers should be ignored in the optimization. But in the completion task, we aim to make the outputs of the network the same as the ground truth ones. Is ignoring the outliers reasonable?

5)	The theoretical analysis is difficult to understand. How is Eq. (8) derived from the analysis in Sec. 3.2? Since the Landau Distribution is not widely used, there should be an introduction about it in the appendix.

6)	Figure 3 seems to be stretched, exhibiting deformation in the text within the figure.

7)	The experiments are problematic. In Figure 4, Tables 1, 2, and 3, there is no comparison with DCD [1] and HyperCD [2], so these results cannot show the superiority of the proposed method. In Figure 4, the “Car” seems worse than the baseline methods.

8)	Are the completed point clouds applied in some downstream tasks, such as classification, part segmentation, and reconstruction?

[1] Density-aware chamfer distance as a comprehensive metric for point cloud completion.

[2] Hyperbolic chamfer distance for point cloud completion.

---

### Official Review · Reviewer_GMc1 · 2023-11-06

**Soundness:** 3 good
**Presentation:** 3 good
**Contribution:** 2 fair
**Rating:** 5
**Confidence:** 4

**Summary:**

The fundamental discrete representations, namely point clouds, serve various fields such as computer vision and robotics. Chamfer Distance (CD) is widely used as a metric and training loss to measure the similarity between two point clouds. However, traditional CD has issues regarding sensitivity to outliers, allowing a few widely distributed points to excessively influence the final similarity score. Additionally, it calculates the average distances of matched point pairs without considering the underlying point-wise distance distribution or addressing differences in pair weights.
In response to these challenges, comprehensive analyses were conducted to prioritize short and long-distance pairs using Gaussian distributions obtained through grid search. Following these findings, an indirect approach identified the Landau distribution, among various distributions, as fitting the bimodal Gaussian mixture model, balancing two types of pairs. Based on these observations, LandauCD was introduced as a new loss function grounded in the Landau distribution.
Through extensive experiments involving LandauCD, significant and consistent improvements were observed across popular baseline networks trained with CD-based losses. These enhancements led to the achievement of new state-of-the-art performance levels on benchmark datasets such as PCN, Shapet-55/34, and ShapeNet-Part. Additionally, the consistent improvements from LandauCD were supported by theoretical explanations.

**Strengths:**

(1) LandauCD is introduced as a regularized Chamfer Distance (CD) loss by integrating the Landau distribution. This methodology aims to reduce the influence of outliers while retaining well-matched point pairs.
(2) The research involves an analysis of the impact of re-weighting functions on distance distributions of point pairs. It provides a systematic approach for selecting a promising re-weighting function based on the discussed mechanism in the study.
(3) Comprehensive experiments are conducted for point cloud completion, resulting in the achievement of state-of-the-art performance on various benchmark datasets.

**Weaknesses:**

(1) Did you compare with Earth Mover's Distance (EMD)? What is the difference between LandauCD and EMD?
(2) In Table 1, was LandauCD utilized in experiments involving FBNet? In Table 2, was LandauCD applied in the experiments conducted with PFNet?
(3) In Table 4, are there any ablation studies on the parameter μ and β?

**Questions:**

My initial rating is 5, since I have some concerns which are illustrated in Weaknesses. I hope the authors could response to those questions in the rebuttal. Then I'll make the final decision.

---

### Official Review · Reviewer_1Bi4 · 2023-11-10

**Soundness:** 2 fair
**Presentation:** 2 fair
**Contribution:** 2 fair
**Rating:** 3
**Confidence:** 4

**Summary:**

The paper proposes a variation of Chamfer distance for point cloud completion. It focuses on understanding the point cloud in a distribution way. Specifically, the similarity of the distribution of the paired point should be considered in the point cloud metrics. Unlike previous loss functions, such as CD and other CD variants, the proposed LandauCD considers both the near-distance and far-distance point pairs. The proposed loss takes more consideration of larger distance values to prioritize the long-distance point pairs by using Landau distribution to approximate a re-weighting function. In a way, the proposed loss function can (1) dynamically re-distribute paired distance in the paired point sets; and (2) make use of both near and far points. Experiments on synthetic datasets ShapeNet show promising results when plugging in different point cloud completion models.

**Strengths:**

- Chamfer distance is a widely used loss function in the 3D point cloud and geometry learning. An improvement to such loss function may have a significant impact on the 3D vision community. The proposed method considers the usually neglected part in the CD loss---the long-distance point pairs, which is interesting and worth exploring.

- The proposed method considers the 3D point in a distribution view. Instead of directly solving for the weighting term, Landau distribution is proposed to approximate the re-weighting function for the point distance. In a way, the re-weighting term is no longer a static scalar, but a dynamically-changed function.

**Weaknesses:**

- A common strategy to deal with the outliers is to remove point distance values larger than some thresholds. I wondered if some comparisons to this truncated CD loss could be presented to further validate the effect of prioritizing long-distance point pairs.

- Is the searched Gaussian distribution general to different datasets? The authors could add experiments.

- Figure 2 does not show a clear pattern of whether the large point distance is reduced when prioritizing the long-distance point pairs.

- I wondered if some theoretical explanation on choosing Landau distribution to approximate the re-weighting function in eq.4 could be presented.

- All these loss functions, including NeurIPS 2021 paper Density-aware CD, are tested on ModelNet, ShapeNet, etc. However, they lack practical use in the real world. I wondered if the authors could provide experiments on some real-world datasets.

- Experimental validation of using two composed distributions over using one distribution (on near or far-distance point pairs) should be provided to further validate the effect of the proposed method.

- As the paper proposed a general loss function, I wondered if some broader use of the loss function could be discussed in the paper.

**Questions:**

I have detailed comments above in the weakness section. I hope the authors could consider improving the paper by adding more theoretical and empirical evidence to validate the effect of the proposed method.